# Optimism and Intolerance to Uncertainty May Mediate the Negative Effect of Discrimination on Mental Health in Migrant Population

**DOI:** 10.3390/healthcare11040503

**Published:** 2023-02-08

**Authors:** Alfonso Urzúa, María José Baeza-Rivera, Alejandra Caqueo-Urízar, Diego Henríquez

**Affiliations:** 1Escuela de Psicología, Universidad Católica del Norte, Antofagasta 1270709, Chile; 2Departamento de Psicología, Universidad Católica de Temuco, Temuco 4780000, Chile; 3Instituto de Alta Investigación, Universidad de Tarapacá, Arica 1000000, Chile

**Keywords:** migration, mental health, depression, anxiety, optimism, intolerance to uncertainty

## Abstract

(1) Background: Migration causes effects on the people who migrate and on the societies that receive them, which can be positive or negative, depending on the characteristics of the interaction. One negative effect is the emergence of mental health disorders associated with the presence of discrimination, a relationship for which there is abundant evidence, although there is less research on factors that may alter this effect. (2) Objective: To evaluate the possible mediating role of optimism and intolerance to uncertainty in the relationship between discrimination and mental health. (3) Method: Nine hundred and nineteen adult Colombian migrants residing in Chile, 49.5% were men and 50.5% women, ages from 18 to 65 years, were evaluated. The Discrimination Experience Scale, BDI-IA Inventory, BAI, LOT-R and the Intolerance to Uncertainty Scale were applied. The effects were estimated using structural equation modeling. (4) Results: A mediating effect of both dispositional optimism and intolerance to uncertainty on the relationship between discrimination and mental symptomatology was observed. (5) Conclusions: The impact on individual suffering and the social cost of mental health problems require investigating variables on the relationship between discrimination and mental health, including mediators of this relationship, which turn out to be central elements in the development of future strategies for the reduction of anxiety and depression symptoms.

## 1. Introduction

The International Organization for Migration (IOM) defines a migrant as a person who moves away from his or her place of usual residence, whether within a country or across an international border, temporarily or permanently, and for a variety of reasons. Migration is a globalized phenomenon that has a high impact on various social areas. It was estimated that, by 2020, more than 281 million people were not living in their country of origin [1].

The migration process involves changes both at the level of the person who migrates and the society that welcomes them, both being in a continuous process of interactions. In this context, a phenomenon that can arise in the receiving society and that has a profound effect on the migrant is discrimination, understood as the different treatment of a group with common characteristics or of a person belonging to that group [2].

In the scientific literature, it is possible to find several reviews that provide evidence of the negative impact of discrimination on health and well-being [3,4,5,6,7,8,9,10,11]. In the migrant population, the two most studied types of discrimination are those of racial and ethnic origin, always associated with poorer health, both physical and mental, as well as lower levels of well-being [12,13,14,15,16,17,18,19,20,21,22,23].

Despite the abundant evidence of the negative effect of discrimination on migrants, there is less frequent research aimed at identifying variables that could influence this relationship and thus reduce the impact of discrimination on health and well-being. In this line, our research group has provided evidence on the mediating role of acculturation stress [24], ethnic identity [25], self-esteem [26,27] and affect [28] on the discrimination-health/well-being/quality of life relationship, which may influence the strength of the relationship.

In the present study, we wanted to investigate the possible mediating role of the variable’s optimism and intolerance to uncertainty in the relationship between discrimination and mental health.

Dispositional optimism can be defined as a facet of personality that is inherently cognitive in nature and relates to the degree to which people have generalized favorable expectations about the future [29,30]. There is abundant evidence on the positive relationship of this trait with good physical health [31] and inversely with the presence of mental health disorders [32,33]. In a similar perspective to the present study, the possible mediating/moderating role of optimism has been studied in other populations without conclusive results. In the LGBTQ+ population, the possible mediating role that the presence of positive (optimistic) beliefs in the family system could have on the negative effect of discrimination on mental health was investigated, but no significant effect was found [34]. A similar result was reported in the Australian Aboriginal population [35], in whom no significant effect of optimism on the racism/mental health relationship was found.

Intolerance to uncertainty can be understood as the excessive tendency of a person to perceive uncertain, unexpected and ambiguous situations as negative, unpleasant, exhausting, unacceptable and threatening, no matter how small the probability of their occurrence. A person with intolerance to uncertainty reacts to these situations in an intense and negative emotional, cognitive and behavioral way and also tends to avoid them [36,37].

Research results using this variable support its association with the presence of mental health disorders, especially anxious and depressive disorders [38,39,40]. Recent studies in the COVID-19 pandemic have provided evidence for the relationship between intolerance to uncertainty and high levels of emotional impact: fears of the coronavirus; sleep problems and emotional symptoms such as worry, stress, hopelessness, depression, anxiety, nervousness and restlessness [41].

Although, in the literature reviewed, it was not possible to find studies relating discrimination and intolerance with uncertainty, evidence has been found on a related, although different, construct, which is the degree of perceived control, considering that at the basis of uncertainty is the perception of a lack of control over what the future holds. In these studies, it has been reported that, in the Arab-American population, a lack of control completely mediated the association between racism and self-esteem and partially between racism and psychological stress [42], while perceived control may mediate the relationship between racism and psychological distress [43]. Similarly, in an Australian Aboriginal population [35], it was found that, in the racism/mental health relationship, having low control may have a mediating effect, while, in an adolescent population, control beliefs significantly mediated the effect of stress on depressive symptomatology [44].

In this context, the aim of this research is to evaluate the possible mediating role of optimism and intolerance to uncertainty in the known relationship between discrimination and mental health, expressed in the presence of anxious and depressive symptomatology. Considering previous evidence, we hypothesized that optimism would diminish the direct effect of discrimination on the presence of anxious and depressive symptoms, whereas intolerance to uncertainty would act by enhancing its negative effect.

This research is framed within the framework of south–south migration, that is, South Americans migrating to countries in the same region. This study deals with Colombians’ migration to Chile, mostly of African descent, which, prior to the pandemic, was characterized by an explosive increase, becoming the first majority of South American migrants before the arrival of Venezuelans. This migratory movement has generated situations of social tension, such as discrimination against migrants, either racially or because of their country of origin, given their link, in the social imaginary, to drug trafficking, drugs and, in the case of women, to the sex trade [24,25,26,27,28].

## 2. Materials and Methods

### 2.1. Sample

This research is a nonexperimental, analytical, cross-sectional study. Non-probability sampling methods were used, combining snowball sampling techniques with purposive sampling for hard-to-reach groups. A total of 919 adult Colombian migrants were evaluated, of whom 476 (51.8%) lived in Antofagasta, 219 (23.8%) in Arica and 224 (24.4%) in Santiago. In terms of gender, 455 (49.5%) were men and 464 (50.5%) were women. The ages of the participants ranged from 18 to 65 years (SM = 35.27; SD = 9.91). Table 1 shows the sociodemographic characteristics of the participants.

### 2.2. Measures

#### 2.2.1. Discrimination

We used the Discrimination experiences scale proposed by Krieger et al. in Spanish [45]. Participants were asked to respond to the following question: Have you ever experienced discrimination, been prevented from doing something or been hassled or made to feel inferior in any of the following situations because of your race, ethnicity, or color? For each situation (At school? Getting hired or getting a job? At work? Getting housing? Getting medical care? Getting service in a store or restaurant? Getting credit, bank loans or a mortgage? On the street or in a public setting? and From the police or in the courts?), the follow-up question was: How many times did this happen: Never (0 point), Once (1 point), Two or three times (2 points), or Four or more times (3 points). In this application, the scale had a Cronbach’s alpha of 0.89.

#### 2.2.2. Depressive Symptoms

We applied the BDI-IA Inventory, a self-reporting instrument of depressive symptomatology composed of 21 items with 4 response options (0 to 3 points) that generate a score between 0 and 63 points. This questionnaire was developed by Beck [46] and has a Spanish version [47]. This instrument has shown adequate psychometric properties in various populations [48]. The internal consistency for the present study was a Cronbach’s alpha of 0.86.

#### 2.2.3. Anxious Symptoms

The Beck Anxiety Inventory (BAI) [49], composed of 21 items that evaluate common symptoms associated with anxiety disorders, was used. Each item is scored from 0 to 3 points. The Spanish version [50,51] was used in this study. In this application, the internal consistency measured by Cronbach’s alpha was 0.94.

#### 2.2.4. Dispositional Optimism

Dispositional optimism was evaluated through the LOT-R (Life Orientation Test-Revised) [52] in its Spanish version [53]. An overall score is obtained through 6 items that assess both optimism and pessimism. The response format is a Likert type with five points (1= strongly disagree to 5 = strongly agree). High scores on this questionnaire indicate high dispositional optimism. The Cronbach’s alpha for the present study was 0.66.

#### 2.2.5. Intolerance to Uncertainty

The scale developed by Freeston [54], in its Spanish version [55], was used. The scale consists of 27 items. The Spanish adaptation is presented in two factors: uncertainty generating inhibition (cognitive, behavioral and emotional) and uncertainty as bewilderment and unpredictability. The reliability of this scale in the present application, measured by Cronbach’s alpha, was 0.83 and 0.86 for the two factors, respectively.

### 2.3. Procedures

The present research is part of a larger project on the effects of discrimination on South American migrants, which has been reviewed and approved by the Scientific Ethics Committee of Universidad Católica del Norte (Res. 011/2018). The initial participants were interviewed mainly in public institutions such as the Chilean Catholic Migration Institute (INCAMI), Global Citizen-Jesuit Migrant Services, Department of Immigration, Colombian Consulate and health centers, among others. Each of them signed a consent form. The data were coded and analyzed with SPSS-21 software, while Mplus version 7 software was used for the structural model adjustment analyses [56].

### 2.4. Statistical Analysis

First, the effect of discrimination on depression and anxiety was estimated using structural equation modeling (see Figure 1). Once the previous relationship was estimated, a second structural equation model—specifically, parallel multiple mediations—was used to estimate the indirect effects of optimism and intolerance of uncertainty on the relationship between discrimination as a predictor variable and depression and anxiety as the criterion variables (Figure 2).

The analyses were performed using the robust weighted least squares (WLSMV) estimation method, which is robust for non-normal ordinal variables [57]. The goodness of fit of all structural models were estimated using chi-square [χ2] values, root mean square error of approximation (RMSEA), comparative fit index (CFI) and Tucker-Lewis index (TLI). According to the standards recommended by the literature [58], the RMSEA ≤ 0.08, CFI ≥ 0.95 and TLI ≥ 0.95 were considered adequate and indicative of a good fit.

Considering the wide age range and possible differences between men and women, we controlled for the effects of sex and age, as well as years of stay, city of residence and declared phenotype in the analyses.

## 3. Results

### 3.1. Descriptives

Table 2 shows some descriptive statistics of all the variables included in the model.

### 3.2. Structural Equation Modeling

The first structural equation model sought to examine the association between discrimination with depression and anxiety in Colombian migrants residing in Chile. The estimated model presented good fit indices, which allowed us to think that the model is a good representation of the observed relationships (Par = 252; χ2 = 4008.339; DF = 1932; *p* < 0.001; CFI = 0.953; TLI = 0.951; RMSEA = 0.035). Figure 3 shows that discrimination has a positive and statistically significant association of moderate magnitude (b > 0.30) [59] with depression (b = 0.313, *p* < 0.001) and anxiety (b = 0.351, *p* < 0.001). Finally, as expected, depression was positively related to anxiety (b = 0.384, *p* < 0.001). The analysis was controlled for the effects of years of stay, city of residence, sex, age and declared phenotype. Solid paths indicate significant relationship effects.

### 3.3. Parallel Multiple Mediations Model

The mediation model presented goodness-of-fit levels close to the criteria recommended by the literature (Par = 350; χ2 = 6303.677; DF = 3211; *p* = 0.000; CFI = 0.944; TLI = 0.942; RMSEA = 0.033). Therefore, the estimated model is a good representation of the observed relationships (Figure 4).

## 4. Discussion

The aim of this research was to evaluate the possible mediating role of dispositional optimism and intolerance to uncertainty in the relationship between discrimination and mental health, expressed in the presence of anxious and depressive symptoms. In this regard, the two proposed models demonstrated adequate fit indicators. Consequently, a direct relationship between discrimination, anxiety and depression was observed, as well as a mediating effect of both dispositional optimism and intolerance to uncertainty in the relationship between discrimination and mental symptomatology.

In relation to the proposed relationship structure, and as indicated by scientific evidence [3,4,5,6,7,8,9,10,11], higher levels of discrimination were associated with higher levels of anxiety and depression. This is particularly relevant for Chile, since the increase in migratory flows has brought with it a deep social crisis related to high levels of xenophobia and discrimination towards migrants, mainly in the northern macro zone of the country. This situation has been aggravated by the lack of state policies [60,61,62], resulting in higher levels of mental health symptoms, such as anxiety and depression.

Discrimination, in addition to having direct effects on the mental health of migrants, also showed indirect effects through the mediating variables studied. It was observed that, to the extent that migrants experienced more discrimination, they perceived themselves as less optimistic and, consequently, presented more mental health symptoms. Thus, it was found that discriminatory treatment would be linked to less favorable expectations about the future, negatively affecting the levels of anxious symptomatology. Studies have documented the importance of optimism in the lives of people belonging to ethnic minorities [63,64,65], so that its approach to experiences of discrimination becomes a significant strategy to reduce mental health problems, physical health and the use of health systems [66]. Regarding the interaction between dispositional optimism and perceived discrimination, the scientific literature has reported that individuals who experience high levels of discrimination and dispositional optimism may have worse psychological, physiological and behavioral responses to stress than those individuals who are very optimistic and experience little discrimination, as well as a more adverse clinical course and greater utilization of medical services [67]. Therefore, while mental health strategies should include increasing dispositional optimism, they should also address the consequences of perceived discrimination as elements that together impact mental health symptomatology.

In addition, discrimination was found to increase intolerance to uncertainty, and this, in turn, increased anxious and depressive symptomatology in the migrant population. Thus, discrimination experiences increased the excessive tendency to consider the occurrence of a negative event unacceptable [36], the overwhelming perception of ambiguous situations and that an uncertain future is unfair [37]. The antecedents on this variable are scarce in the literature regarding migration processes and have been related to the consequences linked to the COVID-19 pandemic [41], so these results are valuable in that they provide innovative evidence for the development of mental health strategies in migrants, considering that the relationship between discrimination and intolerance to uncertainty also presented the greatest direct effects.

These findings are important, given that research has usually examined the association between discrimination and mental health problems without assessing the mediating variables. In this regard, the results obtained in the present research show the relevance of studying dispositional optimism and intolerance to uncertainty in the relationship between discrimination and mental health problems. Dispositional optimism turns out to be a key variable, since, both at the individual and collective levels, it is a source of resources in the face of adverse events and is related to higher levels of well-being [68,69], while intolerance to uncertainty proved to be a crucial variable in adaptation processes, since these are characterized by demands for adaptation with respect to the future. New lines of research should deepen in aspects related to intolerance to uncertainty as a mediating variable between discrimination and mental health, given its effect as a variable susceptible to be modeled with greater potential in health strategies than personality traits, such as dispositional optimism. Although we have found evidence in previous studies that ethnic discrimination (being Colombian) would have a greater weight than discrimination given by racial phenotype [24], and we have also controlled for the effect of the phenotype in the analyses, it would be interesting to continue exploring the possible differences within Colombian migration in terms of skin color and regions of origin, given the heterogeneity of this human group.

Even though we are aware of the multiplicity of factors that can affect mental health, discrimination being one of them, the present research constitutes a valuable contribution to the understanding of the mediating role of optimism and intolerance to uncertainty in the relationship between discrimination and mental health symptomatology. However, some limitations should be considered. The present study employed a nonexperimental cross-sectional design, so future studies should evaluate the temporal stability of the results obtained through longitudinal designs. Another limitation is the difficulty of establishing probabilistic sampling to ensure greater variability in the characteristics of the participants.

We also believe it is important to consider, in subsequent research, further critical reflection on some of the variables used, since a cross-sectional or even longitudinal measurement is not in itself capable of describing post-migratory phenomena such as the interaction between the receiving society and the people who seek to settle in it, including the responses to settlement, such as discrimination, which also has intersectional characteristics, as in this case with the intersection between belonging to a certain nationality (being Colombian) and having a specific skin color.

The exposure of migrant groups to discrimination, the impact on individual suffering and the social cost of mental health problems require investigating new variables regarding the relationship between discrimination and mental health, including mediators of this relationship, such as dispositional optimism and intolerance to uncertainty, which have turned out to be central elements in the development of future strategies for the reduction of anxiety and depression symptoms.

## Figures and Tables

**Figure 1 healthcare-11-00503-f001:**
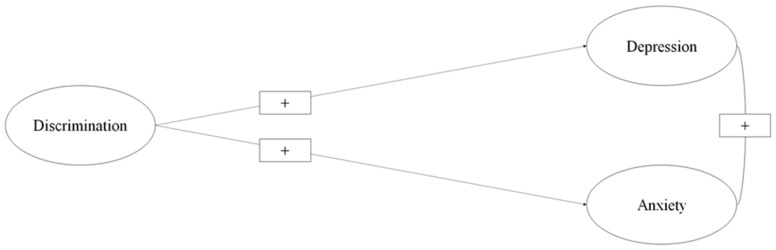
Effect of discrimination on depression and anxiety.

**Figure 2 healthcare-11-00503-f002:**
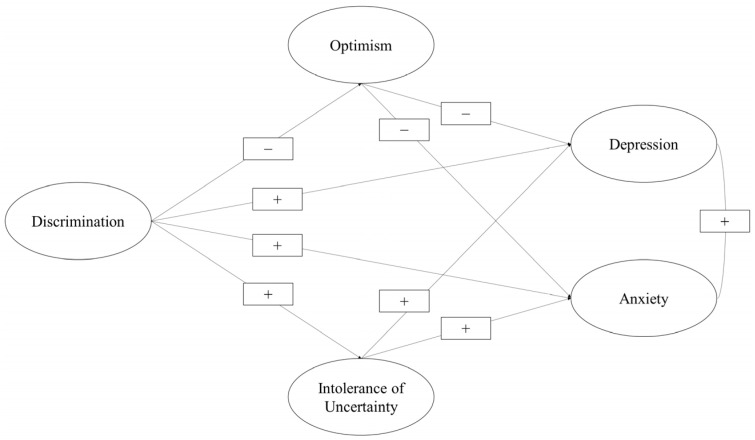
Effects of optimism and intolerance of uncertainty on the relationship between discrimination and depression and anxiety.

**Figure 3 healthcare-11-00503-f003:**
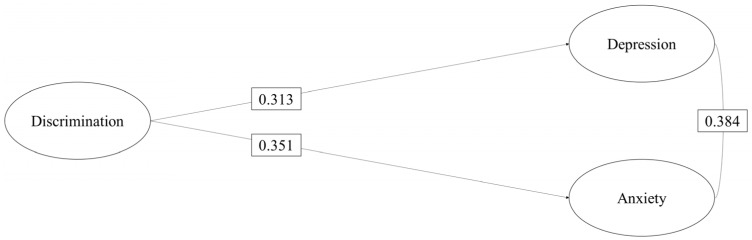
Model of the effect of discrimination on depression and anxiety. All estimated indirect effects were statistically significant (*p* < 0.05).

**Figure 4 healthcare-11-00503-f004:**
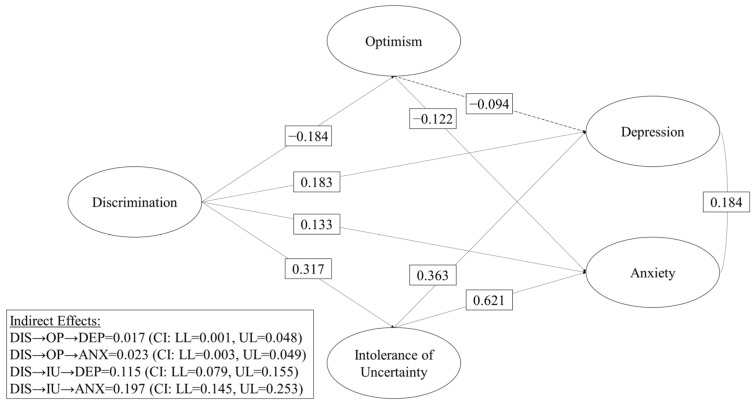
Model of the direct effects of discrimination, mediated by optimism and intolerance to uncertainty on depression and anxiety. Note: DIS = Discrimination; OP = Optimism; IU = Intolerance to Uncertainty; DEP = Depression; ANX = Anxiety; CI = Confidence Interval 95%; LL = Lower Limit; UL = Upper Limit. The analysis controlled for the effects of years of stay, city, gender, age and phenotype. Solid paths indicate significant relationships effects (*p* < 0.05). Non-significant paths are shown with a dashed line. All effects are standardized.

**Table 1 healthcare-11-00503-t001:** Sociodemographic characteristics of participants.

Variables	n (%)
Years of arrival in Chile >10 years 1–10 years Did not respond	40 (4.4) 854 (92.9) 25 (2.7)
Education Incomplete primary education Primary Education Secondary Education Incomplete technical education Technical level Incomplete University education University education Postgraduate Does not respond	102 (11.1) 233 (25.4) 309 (33.6) 82 (8.9) 116 (12.6) 37 (4.0) 20 (2.2) 6 (0.7) 14 (1.5)
Legal situation With residence Visa Without residence Visa Nationalized Does not respond	681 (74.1) 117 (12.7) 61 (6.6) 60 (6.5)
Employment Employee Retired Unemployed Housewife Student Does not respond	656 (71.4) 4 (0.4) 122 (13.3) 59 (6.4) 33 (3.6) 45 (4.9)
Monthly income <125 USD 126–375 USD 376–750 USD 751–1250 USD 1251–1875 USD >1876 USD Did not respond	112 (12.2) 331 (36.0) 355 (38.6) 83 (9.0) 8 (0.9) 7 (0.8) 23 (2.5)
Self-reported phenotype White Indigenous Mestizo Afro descendant Mulatto Others Does not respond	197 (21.4) 38 (4.1) 219 (23.8) 216 (23.5) 161 (17.5) 14 (1.5) 74 (8.1)

**Table 2 healthcare-11-00503-t002:** Scores of the variables included in the model.

Variables	n	ME	SD	OP	IU	DEP	ANX
DIS	781	0.50	0.57	−0.03	0.27 *	0.25 *	0.32 *
OP	876	2.42	0.67		−0.04	−0.02	−0.03
IU	871	2.05	0.76			0.40 *	0.63 *
DEP	794	0.29	0.43				0.41 *
ANX	871	0.48	0.55				

Note: DIS = Discrimination; OP = Optimism; IU = Intolerance to Uncertainty; DEP = Depression; ANX = Anxiety; * *p* < 0.05.

## Data Availability

The data presented in this study are available on request from the corresponding author. The data are not publicly available, because the project has state funding and will only be released once the project is finished.

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
