# Peer review of "Optimism and Intolerance to Uncertainty May Mediate the Negative Effect of Discrimination on Mental Health in Migrant Population"

_healthcare, 2023, doi:10.3390/healthcare11040503_

Round 1

Reviewer 1 Report

A definition of the term migrant in the context of the paper is missing, but would be essential for a better understanding of the methodology (description of the assessed group) further analyses and results.

Please elaborate on the correlations/similarities of perceived control and intolerance of uncertainty (p.2), as it is referenced here that it is a similar construct. I agree with this, but ask for an explanatory elaboration.

A subdivision into different age groups and the gender distribution in these, as well as their comparison, would be another very interesting point in the analysis. The range here is very wide. The information on the characteristics of the group covered is also very limited and does not include information on the migration and integration process; the length of stay and residence status are further key factors for mental health being, as different studies also show. Those are mentioned in a note (The analysis was controlled for the effects of years of stay, city of residence, sex, age, and 173 declared phenotype. P. 5), but further information regarding the group covered (representation among the sample, differentation of different groups in the analysis (e.g. according to duration of stay and residence status) are missing.

Ethnicity and race are asked as grounds for discrimination, but there is no more specific information regarding the characteristics of the group covered. It must also be taken into account that the heterogeneity among Colombian migrants is undoubtedly very high (also with regard to ethnicity). Here, too, a differentiation and inclusion in the analysis would be very interesting, or at least a corresponding elaboration under limitations.

Acculturation is mentioned several times (also in the discussion), but it is a complex term that needs an appropriate definition and elaboration of the interrelations with the content of the paper.

The framework conditions (society, migration governance, etc.) play a major role in migration and integration processes, the perception of migration/migrants among the population and the causes and perceptions of discrimination and are not sufficiently elaborated except for a few sentences in the Discussion. Even if the focus of the paper is on statistical analysis, the contextuality of the data and results must be sufficiently included.

An assessment of the method, even taking into account the information provided, is made much more difficult because the questionnaire itself is not available. However, this would be very relevant in order to be able to assess the actual implementation and further analysis. The description of the results and the figures are there, however, an improvement of the comprehensibility could be achieved by more extensive descriptions. This would also improve the comprehensibility of the reference to the statements in the discussion.

In general: The relevance is high and results can indeed contribute significantly to the design of health interventions for migrants. However, it must be taken into account that the factors influencing (mental) health of migrants are highly multifactorial, as different research also demonstrates. In order to avoid a simplification of this complex issue, a corresponding paragraph in the text would be useful.

Author Response

Thanks for your suggestions

Reviewer 2 Report

I thank the authors for sharing their work. The research is very interesting, its methodology is clearly presented and the graphics are enhancing the understanding of the results. However, there are some conceptual and contextual aspects that remain missing. As stated in the suggested improvements below, the authors should improve these parts of the paper before its publication.

suggested improvements:

- In the introduction (1st sentence) the authors should clarify what migration they are talking about. De Haas, Castles and others have demonstrated that internal migration is far outnumbering international migration. The study discusses the situation of border crossers so that should be clearer, also in order to avoid reproducing migration discourses that do not reflect different forms of human mobility. Accordingly, further statements on migratory behavior and its effect on people's lives in the text should be adapted. 

- source 2 is insufficient as a scholarly background on describing post-migratory phenomena such as the interaction between receiving society and people who seek settlement therein, including responses to settlement such as discrimination. I highly recommend considering Castles' "Age of Migration" for an overview study, and Zapata-Barrero and Vertovec considering intercultural societies and discriminatory phenomena. 

- the definition of discrimination is insufficient in so far that it does not detail what is implied in the notion of "common characteristics". The authors should consider intersectionality theory to clarify. 

- The notions of racial and ethnic discrimination need defining. What do the cited studies understand by these notions and how do the authors of this study employ the terms?

- line42: the authors mention abundant evidence of discrimination experiences by migrants, but no evidence is presented in this spot. Consider international studies (FRA for example).

- lines 45-46: the terms acculturation stress, ethnic identity, self-esteem and affect need to be clearly defined to make clear how they were used by the research group in the cited previous work and the current paper. This goes for all terms used in this paper without further clarifying what is meant.

- lines 49-57: The research question needs clarifying as it is not clear what population was researched, nor has it be clarified why the authors have chosen this perspective and a grouping of the named aspects for the study. while the authors demonstrate definitions and specific examples, the way in which they are presented needs restructuring to provide readers sufficient background before making claims.

- lines 64-68: This sentence is too dense and needs unpacking.

- line 78: what is the gender dimension of the study on the Arab-American population? 

- line 81: what is the gender dimension in the Australian Aboriginal population?

- lines 169-170: the author's mention expectations prior to data evaluation but the hypothesis is not clearly articulated in earlier parts of the paper- Perhaps it would have made sense to state that in the discussion clearly, or even earlier.

- it is only in the discussion that the authors introduce contextual factors such as the situation of migrants (even though unclear how that experience is gendered, class-distinct, age-distinct). That should be introduced much earlier so that the discussion can be used as a space to evaluate the sampled data against the backdrop of contextual factors.

- contextual factors in Chile need clarification: what is meant by migratory flows? what is the social crisis stated, where is it stated, and how do the authors know that it affects both migrant and non-migrant population in Chile? Specifically, it would make sense determinign what kind of state policies were relevant. The notion of acculturation processes is contested in problematized in the literature in migration studies. I suggest the authors consult this body of scholarship to clarify what this means in the context of Chile in terms of expectations for migrants and the subsequent effects on mental health.

- the study body does not focus on women exclusively, which is what the abstract suggests so I suggest that the authors adapt to include the almost equal distribution of men and women participants.

Author Response

Thanks for your suggestions

Reviewer 3 Report

The introduction is to general. It should be more specific, especially stating clearly who the subject matter is. This only comes in the method section that mention Columbian migrants living in Chile. There is no context provided of why they are there. Thus, it just glosses over who are being studied. The research does not provide specific information but just continues along the line of general findings. There needs to be specific information added. 

There are several grammar mistakes that needs to be corrected for instance from the very beginning in line 11—bakground should be Background. The sentences tend to be run-ons and the structure are not clear. 

A definition of dispositional optimism should be provided. This would help clarify some of the statements made and also help draw conclusions to the findings in the research. For instance, lines 49-52 is unclear. It really doesn’t provide what is the aim of the study. What is the former and what is latter? The assumption that there is a direct cause and effect should be re-evaluated. Is the discrimination on the symptoms or is it on the person?

“In the present study, we wanted to investigate the mediating role of the variables of optimism and intolerance to uncertainty, under the hypothesis that the former would diminish the direct effect of discrimination on the presence of anxious and depressive symptoms, while intolerance to uncertainty would act by enhancing its negative effect.”

 Thus, it was very difficult to read through this current paper. 

Author Response

Thanks for your suggestions

Round 2

Reviewer 1 Report

With the consideration of the changes made and the answers to the reviewer report, the paper can be published from my point of view.

Reviewer 3 Report

This version is better. The terms were clarified and more substantive discussion was given. The study can be expanded and go further in depth. But this version is adequate.